# Theoretical Study of Si/C Equally Mixed Dodecahedrane Analogues

**DOI:** 10.3390/molecules28062769

**Published:** 2023-03-19

**Authors:** Tamotsu Uchiyama, Taiji Nakamura, Miyabi Hiyama, Takako Kudo

**Affiliations:** 1Department of Environmental Engineering Science, Graduate School of Science and Technology, Gunma University, Kiryu 376-8515, Japan; 2Institute for Materials Chemistry and Engineering, Kyushu University, Nishi-ku, Fukuoka 819-0395, Japan; 3Gunma Study Center, The Open University of Japan, Maebashi 371-0032, Japan

**Keywords:** dodecahedrane, Si substitution, strain energy, relative stability, regression analysis, half silicon substitution, ab initio molecular orbital calculation

## Abstract

To gain insight into the effect of Si/C arrangement on the molecular framework of strained polyhedral compounds, dodecahedrane analogues containing equal numbers of carbon and silicon (Si/C equally mixed dodecahedrane analogues) were investigated using the ab initio molecular orbital method. There are 1648 isomers for which the Si/C arrangement on the molecular framework is different. Based on the geometry optimization of all the isomers as well as the carbon and silicon analogues, the characteristics of the structure, relative energy, and strain energy of the Si/C equally mixed compounds are presented. Then, important factors controlling the relative energy, such as strain energy, are proposed through regression analysis. Also discussed is the correlation between the relative energy and the indices of Si/C dispersion, such as the number of skeletal C–Si single bonds and condensed five-membered rings constituting the polyhedral structure.

## 1. Introduction

Tetrahedrane, cubane, and dodecahedrane are known as saturated hydrocarbons with the structure of a regular polyhedron of Plato [1]. These compounds have long been an attractive subject for many experimental and theoretical studies because of scientific curiosity about their highly symmetric structure and the significant strain energy brought about by the deformed sp^3^ structure. Every polyhedral compound has already been synthesized. Tetrahedrane has been synthesized as derivatives with bulky substituents, while cubane and dodecahedrane have been obtained as genuine hydrocarbons, C*_X_*H*_X_*(*_X_* = 8,20) [2,3,4,5,6,7,8,9,10]. Dodecahedrane, the target of the present study, has a highly symmetric structure (the point group of C_20_H_20_ is *I_h_*) and is known as the completely hydrogen-saturated dodecahedral fullerene, which is the smallest fullerene [11,12]. Persiladodecahedrane (also called *persilafullerane*), the structure of which encapsulates a Cl^−^ anion, has recently been synthesized by Bamberg et al. [13,14]

Among the polyhedral compounds, tetrahedrane and cubane are highly strained due to the condensed three- or four-membered rings in their molecular skeletons. The strain energy of dodecahedrane, however, is remarkably lower than those of tetrahedrane and cubane as the molecule’s structure is constructed from less strained five-membered rings [15]. Paquette’s group has experimentally determined the strain energy of dodecahedrane to be 61.4 kcal mol^−1^ [16,17], which is significantly smaller than the strain energies of tetrahedrane (140.0 kcal mol^−1^) and cubane (154.7 kcal mol^−1^) [18].

For the theoretical studies, Nagase et al. suggested that a small polyhedral hydrocarbon’s strain energy decreases with the substitution of the skeletal carbons by the heavier group 14 elements, especially for the cages containing four–membered rings, through the ab initio molecular orbital method [19,20]. Earley systematically investigated the strain energy of 10 types of polyhedral compounds, and the heavier group 14 elements substituted analogues by ab initio molecular orbital calculations [15]. The author suggested that the strain energies of polyhedral compounds are affected by the inner condensed ring’s strain, and the former can be approximately assessed as the sum of the latter.

Recently, compounds with a structure containing both carbon and silicon in the molecular skeleton have attracted attention. For Si/C mixed unsaturated compounds, Si-doped fullerene analogues in which the skeletal carbons are substituted by silicons have been confirmed by photoionization mass spectroscopy [21,22,23]. Furthermore, we revealed that some of the C and Si alternately mixed annulenes (benzene and cyclooctatetraene analogues, and so on) have unique characteristics that are completely different from those of the carbon and silicon congeners [24,25,26]. On the other hand, silaadamantanes and silicon carbides, both of which are known as functional materials, are the typical examples of the Si/C mixed saturated compounds. Fotooh et al. investigated the HOMO–LUMO gap changes brought about by Si substitutions in adamantane [27]. Miranda et al. obtained the thermodynamic quantities of silaadamantane with theoretical calculations and compared them with those of adamantane and persilaadamantane [28]. Silicon carbides with diamond-type crystal structures made of only C–Si bonds have been utilized in a wide variety of fields because of their unique semiconductor and mechanical characteristics. They are expected to be the next generation of power semiconductors of the.

For saturated cyclic or polyhedral compounds, the Si/C mixing effect on the angle strain, which is one of their important characteristics, is interesting. However, there are only a few theoretical studies for two-element-mixing analogues of tetrahedrane and cubane [29,30]. In this situation, our fragment structure energy (FSE) analysis demonstrated that Si/C alternately mixing contributes to the stabilization of the systems in the ideal (with no angle strain) sp^3^ environment as well as the sp^2^ environment, which was not observed for the mixing of carbon or the other heavier group 14 elements [31].

The goal of the present study was to explore the Si/C mixing effects on the structure and properties such as strain energy of polyhedral compounds through quantum chemical calculation. For the purpose, we decided to investigate the Si/C equally mixed dodecahedral analogues, C_10_Si_10_H_20_. The main reason is that the number of the isomers with different Si/C arrangements is the largest [32,33,34], 1648, among a series of Si/C mixed dodecahedrane analogues, C*_X_*Si_20–*X*_H_20_(*X* = 0,1,2, … ,20), so the trend in the effect based on large amounts of data is considered to be rather reliable. First, the structure of all the isomers was determined, and then the relative energy and strain energy were compared not only for the Si/C mixed analogues but also for dodecahedrane (C_20_H_20_) and persiladodecahedrane (Si_20_H_20_). In addition, regression analysis was carried out to examine the correlations between the relative energy, strain energy, and indices of the Si/C arrangement.

## 2. Methods

The geometries of all molecules were fully optimized at the Hartree–Fock (HF) [35] and the second-order Møller–Plesset perturbation (MP2) [36,37] level with the 6-31G(d) basis set [38]. Then, normal mode vibrational analysis was performed to confirm the molecular characteristics as an equilibrium structure. As a result, all molecules investigated here were found to be equilibrium structures at the HF/6-31G(d) level. For some specific molecules, the analysis was carried out at the MP2/6-31G(d) level as well, and they were confirmed to be equilibrium structures at both levels of calculations.

Strain energy (*SE*) was estimated from the homodesmotic reaction energy (∆*E*), described by the following equation [39,40]:(1)$$\begin{array}{c} C_{10}{\mathrm{Si}}_{10}H_{10}+n_{\mathrm{CC}}C_{2}H_{6}+n_{\mathrm{CSi}}CSiH_{6}+n_{\mathrm{SiSi}}{\mathrm{Si}}_{2}H_{6}\\ =\overset{3}{\underset{a=0}{\sum }}\underset{A=C,\mathrm{Si}}{\sum }n_{aA}{\left({\mathrm{CH}}_{3}\right)}_{a}{\left({\mathrm{SiH}}_{3}\right)}_{3–a}\mathrm{AH}+\Delta E \end{array}$$

C_10_Si_10_H_20_ is the energy of Si/C equally mixed dodecahedrane analogues, while C_2_H_6_, CSiH_6_, and Si_2_H_6_ are, respectively, the energies of ethane, methyl silane, and disilane. On the other hand, *n_CC_*, *n_CSi_*_,_ and *n_SiSi_* are, respectively, the numbers of C–C bonds, C–Si bonds, and Si–Si bonds. The first term of the right side of Equation 1 represents the energy of Si/C mixed isobutane analogues that comes from each vertex atom and the three atoms connected with that atom. Therefore, the *n_aA_* coefficients are different for every isomer because they depend on the arrangement of Si and C atoms in the framework, and *n_CC_*, *n_CSi_*, and *n_SiSi_* are determined from *n_aA_.* The details are in the Appendix A.

Furthermore, single- and multiple-regression analyses were carried out to assess the contribution of crucial factors controlling the relative stability of the isomers. All calculations were performed with the Gaussian 16 program package [41].

On the other hand, 1648 Si/C arrangements of the isomers were obtained with our own program. The details are presented in the Appendix A.

## 3. Structures

Seven isomers (***a***–***g***) with different Si/C arrangements are depicted in Figure 1. Among the 1648 isomers, they are especially characteristic for their relative energy (*RE*), strain energy (*SE*), and the number of C–Si bonds (*N_C–Si_*).

The properties are compared among all the 1648 isomers together with some other items in Table 1. Isomers ***a***–***d*** have 24 C–Si bonds, which is the largest *N_C_*–*_Si_* among all the isomers. The dodecahedral framework consists of 30 sides (bonds), so for ***a***–***d***, 80% of the bonds are C–Si bonds. In contrast, the *N_C_*–*_Si_* of ***g*** is the smallest, as it is only six. As *N_C_*–*_Si_* is considered to be an index of dispersion of C and Si in the molecular framework, the two elements are considerably delocalized in ***a***–***d***, while they are extremely localized in ***g***, which is obvious from the figure.

The isomer ***a*** has the lowest electronic energy at the present level of theory. Therefore, *RE* is the energy of each isomer relative to that of ***a***. As shown in Table 1, however, it is almost the same for ***a***–***d***. In fact, we found that the order of the relative stability can change at other levels of theory, but the energy differences are still small (see Appendix A). A histogram of the number of isomers from the *RE* is shown in Figure 2. As in the case of ***a***–***d***, it was found that there exist many isomers in between the tiny energy range (5.0 kcal mol^−1^) except for ***f***, suggesting this is one of the characteristics of the Si/C equally mixed dodecahedrane analogues. On the other hand, ***e,*** with the second-largest *N_C_*–*_Si_* (20), has the smallest *SE*, 29.9 kcal mol^−1^. It is noteworthy that the *RE* of ***e*** is also close to those of ***a***–***d***, which have the largest *N_C_*–*_Si_*.

The *SE* of the Si/C mixed isomers is between those of dodecahedrane and persiladodecahedrane but rather close to that of the carbon analogue. Incidentally, the *SE* of the silicon analogue estimated by Earley is much larger than ours, suggesting that the effect of electron correlation might be important for the estimation of the strain energy of the silicon analogues consisting of flexible Si–Si bonds (see Appendix A). The isomer ***f*** has the largest *SE* among all the 1648 isomers, and its *SE* is even larger than that of dodecahedrane. In fact, it was found that in addition to ***f***, there are eight more isomers that have a larger *SE* than dodecahedrane. Obviously, *RE* seems to be proportional to *SE* among the Si/C mixed isomers.

Table 2 shows each bond length for the isomers mentioned above. The C–C and Si–Si bond lengths of dodecahedrane and the silicon analogue are also shown for comparison. The calculated lengths of the C–C bonds of C_20_H_20_ and the Si–Si bonds of Cl^−^@Si_20_H_20_ agree well with their respective experimental values. The encapsulated Cl^−^ does not seem to have any effect on the dodecahedral structure. In addition, every bond length except for that of the C–Si bond of Si/C equally mixed isomers is not much different from those in C_20_H_20_ and Si_20_H_20_. The C–Si bond lengths of ***a****–**e*** are close to those of methylsilane CH_3_SiH_3_ (1.88 Å) obtained at the same level of theory, suggesting the bond length is almost not influenced by the polyhedral structure. On the other hand, the corresponding bond lengths of ***f*** and ***g*** are longer than those of the other isomers, so the relatively large strain energy might affect the C–Si distance and relative stability.

Finally, the HOMO–LUMO energy gap of the dodecahedrane analogues was examined. For comparison, the energy gap of diamond-type molecules with molecular sizes similar to those of dodecahedranes is also shown in Table 3 as the model of silicon carbide. C_30_H_40_ and Si_30_H_40_ are the model compounds of the diamond-type crystal terminated by hydrogens and the silicon analogue, respectively, while C_10_Si_20_H_40_ and C_20_Si_10_H_40_ are the two types of Si/C mixed congeners [31]. The frontier molecular orbitals of the seven isomers ***a*** to ***f***, dodecahedrane, and persiladodecahedrane are displayed in Appendix A. As the table shows, the energy gaps of the Si/C mixed analogues are halfway between those of dodecahedrane and persiladodecahedrane, except for isomer ***f***. This can be explained by the fact that the frontier MOs of ***f*** tend to have large coefficients, especially on the long Si–Si σ-bond sequence (see Appendix A), so the energy gap becomes close to that of σ and σ* of Si–Si bonds in Si_20_H_20_, which is known as theσ-conjugation of Si–Si bonds. However, a surprising thing is that the gap is rather smaller than that of the silicon analogue. Incidentally, there are eight more isomers whose HOMO–LUMO gap is smaller than that of Si_20_H_20_. Long Si–Si bond sequences such as those in isomer ***f*** are also seen in their structures (see Appendix A). An interesting thing is that the *SE* of these isomers is larger than that of dodecahedrane, as mentioned above. The HOMO–LUMO gap of isomer ***g*** with the branched Si–Si bonds is also small compared with that of the other isomers, probably for the same reason.

For the model molecules of silicon carbide, the observations were similar to those for the dodecahedrane analogues, though the energy gap of the Si/C mixed dodecahedranes is about 1 eV smaller than the average of the two types of the Si/C alternately mixed diamond-type molecules (12.9 eV, see Table 3). The small HOMO–LUMO energy gap can be considered to be one of the requirements of a molecule as a good semiconductor. In this sense, some isomers of the Si/C equally mixed dodecahedranes were found to have the potential for such functional materials.

As a result, Si/C mixed dodecahedrane analogues such as isomer ***f*** might be semiconductor candidates depending on the Si/C arrangement in their molecular framework. Moreover, this is an interesting example of a Si/C arrangement significantly affecting a molecular property.

## 4. Regression Analyses for the *RE* of Isomers

In the previous section, it was revealed that some important factors such as strain energy (*SE*) and the number of C–Si bonds (*N_C_*–*_Si_*) affect the relative energy (*RE*) of every isomer. Therefore, we tried to determine how important each factor is for the relative stability of all the 1648 isomers using regression analyses.

### 4.1. Relative Energy vs. Strain Energy

Figure 3 shows the correlation of *RE* against *SE*. The correlation coefficient (R) and the coefficient of determination (R^2^) are, respectively, 0.9432 and 0.8897, suggesting a considerably strong correlation between *RE* and *SE*. Therefore, regarding the strain energy, the Si/C equally mixed dodecahedranes seem to have a common characteristic as strained polyhedral compounds. Furthermore, it is obvious from the figure that isomer ***f***, with the largest *SE* as well as *RE* among all the isomers, is specifically unique.

### 4.2. Relative Energy vs. the Number of C–Si Bonds

As shown in the preceding section, the most stable isomers have the largest *N_C_*–*_Si_*, which means that a structure with delocalized C and Si tends to be relatively stable. However, as seen from the R and R^2^ in Figure 4, there is a weak correlation between *N_C_*–*_Si_* and *RE*, and it is not that high compared with that with *SE*. This result is considered to be evidence showing the superiority of *SE* over *N_C_*–*_Si_* with regard to the impact on the *RE*.

### 4.3. Relative Energy vs. Five-Membered Rings of the Molecular Skeletons

Some correlation between *RE* and *N_C–Si_* was expected from the fact that a group of isomers with the smallest *RE* tend to have the larger *N_C–Si_*. However, according to the regression analysis described in Figure 4, the correlation was found to not be as high as noted above. Therefore, to look at the dispersion of C and Si from another point of view, we tried to focus on the twelve five−membered rings (5MRs) constituting the molecular framework of the Si/C equally mixed dodecahedrane analogues.

As shown in Figure 5, there are eight types of 5MRs depending on the Si/C arrangement of the pentagonal vertexes. The extent of the Si/C delocalization/localization is estimated by the comparison of the two types of C2Si3 (5MR−3 and 4) and Si3C2 (5MR−5 and 6). For both cases, C and Si are more delocalized in the latter than the former, as seen from Figure 5.

Figure 6 shows the correlation between the number of each 5MR (N_5MR−X_; X = 1−8) and RE. For every 5MR, the number is limited because of the restriction on the number of C and Si being 10 each in the dodecahedral framework. For example, as the number of 5MR, 0−3 are only allowed for 5MR−1 and 5MR−8, while those are 0−5 for 5MR−2 and 5MR−7, and so on. Among them, the coefficients of correlation for *N_5MR−3_* and *N_5MR−5_* (“localized” group) are about 0.6, so these two types of 5MR have a weakly positive correlation with *RE*, while there are weakly negative correlations (R ≈ −0.6) between *RE* and 5MR−4,6 (“delocalized” group). These results suggest that isomers consisting of the Si/C delocalized 5MR are relatively stable, though the correlation is not high.

Next, we also examined the correlation between *RE* and the sum of the *N_5MR−X_* of groups consisting of two kinds of 5MR having the same tendency, 5MR−3, 5 (“localized” group) and 5MR−4, 6 (“delocalized” group). As shown in Figure 7a, *RE* has a stronger correlation with the sum of *N_5MR−3_* and *N_5MR−5_* in comparison with that of the respective single 5MR cases mentioned above. Likewise, the sum of *N_5MR−4_* and *N_5MR−6_* has a stronger negative correlation with *RE*, as seen in Figure 7b. Therefore, these results indicate that isomers with the structure in which C and Si are rather delocalized in 5MRs tend to be stable, and the same is true for the number of C–Si bonds shown in Figure 4.

From the investigation in this section, it was found that the specific five−membered rings considerably affect the relative stability of the isomers.

### 4.4. Multiple Regression Analysis of SE and N_C_−_Si_

We examined the three important properties *SE*, *N_C_–_Si_*, and *N_5MR−X_* correlated with the *RE* of the isomers as discussed above. So, our next concern was quantitatively comparing each contribution to *RE*. Apparently, *SE* has a strong correlation, but that of the others is rather mild. Furthermore, *N_C_–_Si_* and *N_5MR−X_* (the sum of *N_5MR−3_* and *N_5MR−5_*, etc.) are not considered to be factors completely independent of each other, as both are related to the dispersion of *C* and *Si* in the molecular framework. Furthermore, both variables have some correlation with *SE*, the coefficient of correlation being −0.6178 for *N_C_–_Si_* and 0.8897 for the sum of *N_5MR−3_* and *N_5MR−5_* (see Appendix A). It is not appropriate to select variables having a strong correlation with each other. Then, we chose *N_C_–_Si_*, with a smaller correlation with *SE*, as an index of the delocalization of *C* and *Si* and performed a multiple regression analysis of *SE* and *N_C_–_Si_*.

The multiple regression equation for this study was as follows:*RE^s^* = *a^s^_SE_*
*SE^s^* − *a^s^*_*C**Si*_
*N^s^_C_*–*_Si_*(2)

*RE*^s^ is a response variable for standardized *RE*. *SE^s^* and *N^s^_C_*–*_Si_* are, respectively, the explanatory variables for standardized *SE* and *N_C_*–*_Si_*; and *a^s^_SE_* and *a^s^_CSi_* are the coefficients, respectively.

The details are provided in the Appendix A.

As a result, the following Equation (3) was obtained.
*RE^s^* = 0.77 *SE^s^* − 0.27*N^s^_C_*–*_Si_*(3)

Based on this analysis, the determination coefficient was confirmed to be extremely high, 0.9363.

The coefficient of *SE^s^* is about three times larger than that of *N^s^_C_*–*_Si_* at the present level of calculations, suggesting that the strain energy is a major factor controlling the relative stability of the isomers of the *Si/C* equally mixed dodecahedrane analogues.

## 5. Conclusions

The effect of half-silicon substitution of the skeletal carbons of dodecahedrane on the structure and some properties, such as strain energy, was investigated through the ab initio molecular orbital method. There are 1648 isomers of the Si/C equally mixed dodecahedrane, in which Si/C arrangement energies are very close. Among them, however, the least stable isomer involving a long Si sequence in the molecular structure was found to have remarkably high relative energy and strain energy. The strain energies and HOMO–LUMO gaps of the Si/C mixed compounds are intermediate between those of dodecahedrane and the per-silicon analogues, except for a few isomers. These isomers seem to be considerably unique among all the Si/C equally mixed dodecahedrane analogues. The relative stability of all the isomers tends to be high when the strain energy is low, and the C and Si are quite delocalized in the dodecahedral framework.

According to the regression analysis, the strain energy was found to be highly correlated with the relative energy. Also examined were the contributions of the number of C–Si bonds and the condensed five-membered rings in the dodecahedral skeleton. Both are expected to have some important effect on the relative energy as they can be the indexes to assess the Si/C dispersion, but the correlation with the relative energy was rather weak. Furthermore, standardized multiple regression analysis revealed that the impact of the strain energy on the relative energy is greater than that of the number of C–Si bonds.

From the present study, the properties of Si/C equally mixed dodecahedranes were found to greatly depend on the Si/C arrangement: the silicon substitution, in other words. Therefore, molecular designs that take the results into account may contribute to the development of a new functional material such as silicon carbide. Investigations of Si/C mixed dodecahedranes with other Si/C ratios are underway.

## Figures and Tables

**Figure 1 molecules-28-02769-f001:**
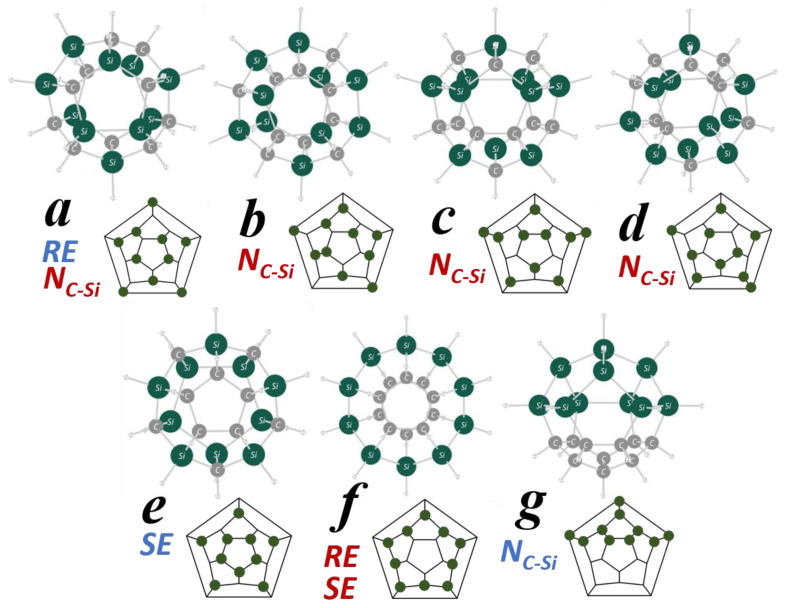
The MP2/6-31G(d) optimized structure and polyhedral graph of seven isomers of Si/C equally mixed dodecahedrane analogues of C_10_Si_10_H_20_. Green circles show Si and gray ones are C in the structures, while green dots represent Si in the graphs. For the *RE*, *SE*, and *N_C–Si_*, red characters indicate that the property is the largest among *all 1648* of the isomers, while blue ones indicate the smallest property.

**Figure 2 molecules-28-02769-f002:**
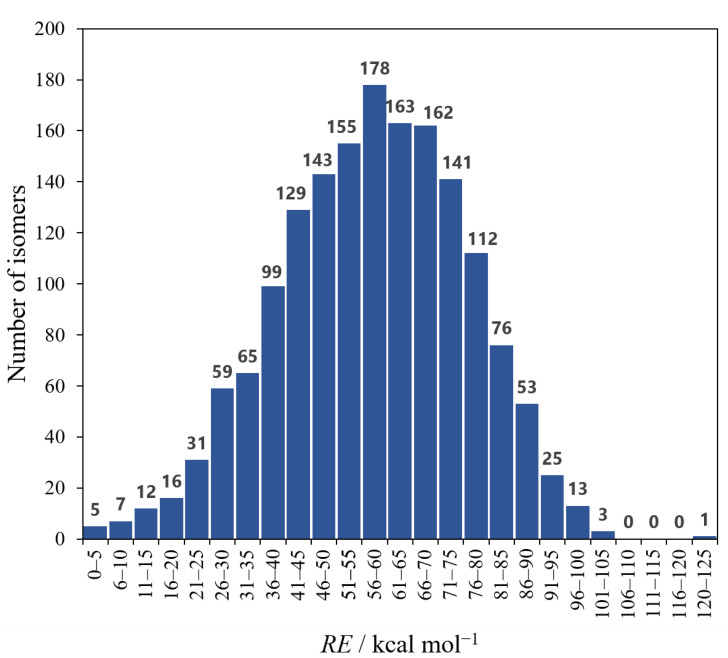
Histogram of the number of isomers and *RE*. The class range of *RE* is 5.0 kcal mol^−1^.

**Figure 3 molecules-28-02769-f003:**
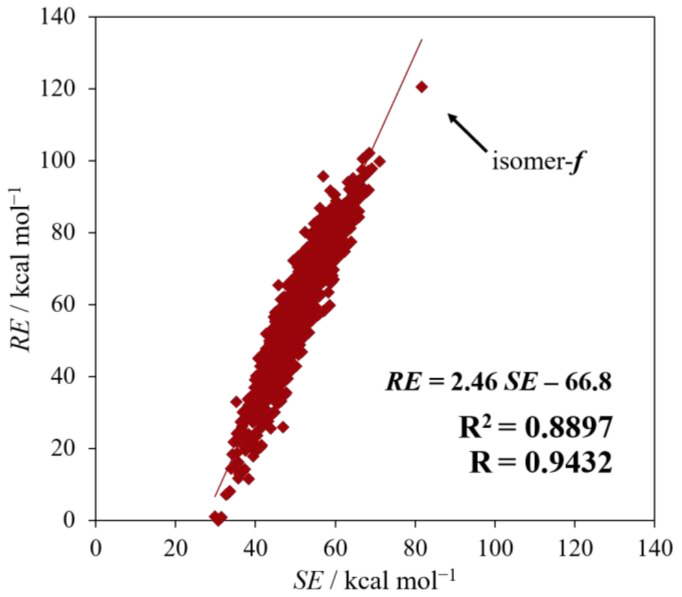
The results of regression analysis between *SE* and *RE*. The arrow indicates isomer ***f***.

**Figure 4 molecules-28-02769-f004:**
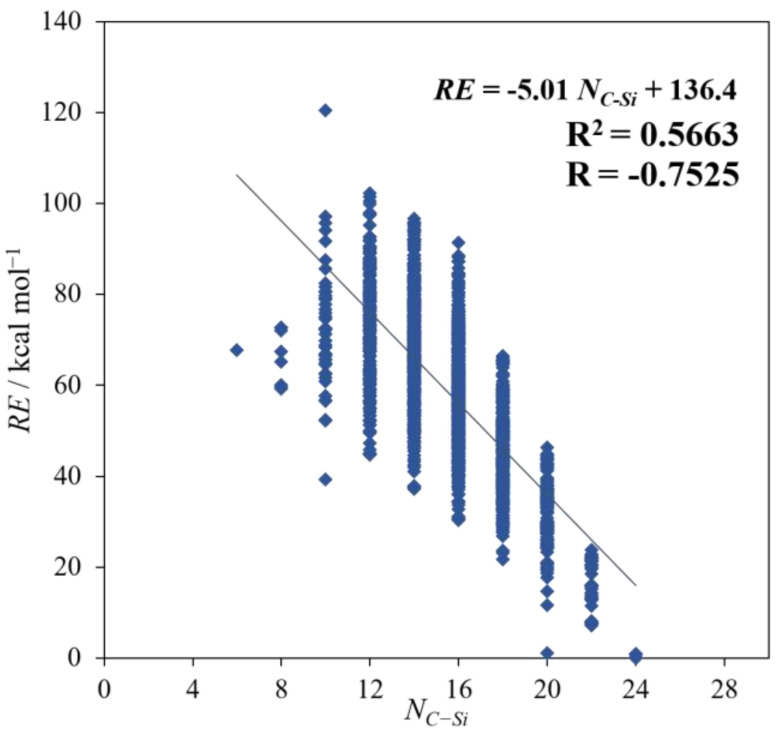
The plots of *RE* against *N_C_*–*_Si_*.

**Figure 5 molecules-28-02769-f005:**
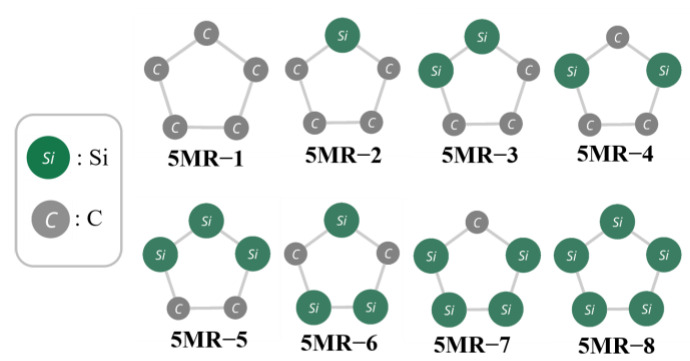
All types of 5−membered rings (5MRs) in Si/C mixed dodecahedrane analogues.

**Figure 6 molecules-28-02769-f006:**
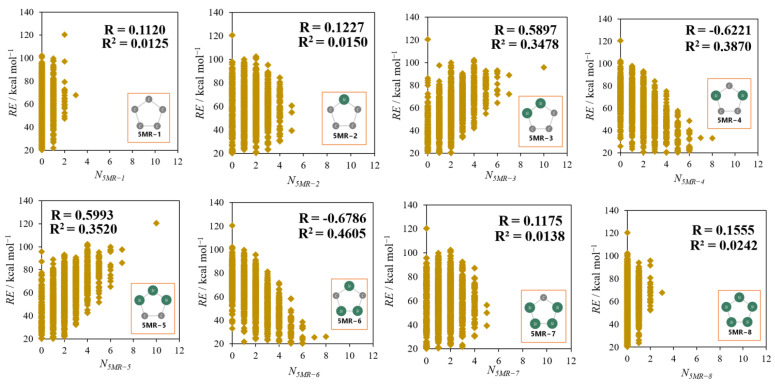
The correlations between *RE* and *N_5MR__−X_* for the 8 types of 5MR.

**Figure 7 molecules-28-02769-f007:**
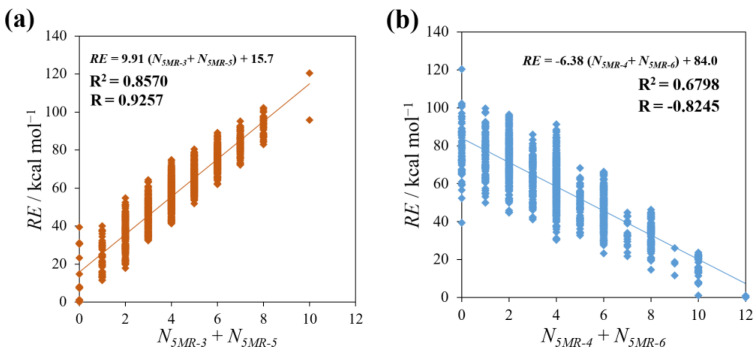
The plots between *RE* and the number of the sum of two 5MRs that have a similar correlation for *RE*: (**a**) 5MR−3 + 5MR−5 (“localized” group) and (**b**) 5MR−4 + 5MR−6 (“delocalized” group).

**Table 1 molecules-28-02769-t001:** Several important properties of the Si/C equally mixed dodecahedrane isomers, dodecahedrane, and the silicon analogues at the MP2/6-31G(d) level.

Isomer	Point Group	C–C	C–Si	Si–Si	*RE* /kcal mol^−1^ [*RE*(+ZPC)] **	*SE* /kcal mol^−1^	Note
* **a** *	*C_2_*	3	24	3	0 [0]	30.6	The lowest electronic energy so *RE* is 0.0, The largest *N_C_*–*_Si_*
* **b** *	*C_1_*	3	24	3	0.1 [0.2]	30.8	The largest *N_C_*–*_Si_*
* **c** *	*C_3v_*	3	24	3	0.8 [0.7]	31.4	The largest *N_C_*–*_Si_*
* **d** *	*C_3_*	3	24	3	0.9 [0.9]	31.5	The largest *N_C_*–*_Si_*
* **e** *	*C_5v_*	5	20	5	1.1 [2.5]	29.9	The smallest *SE*
* **f** *	*D_5d_*	10	10	10	120.5 [123.0]	81.7	The largest *RE* and *SE*
* **g** *	*C_3v_*	12	6	12	67.7 [72.7]	50.4	The smallest *N_C_*–*_Si_*
C_20_H_20_	*I_h_*	30	0	0	–	54.1 (43.7 *)	–
Si_20_H_20_	*I_h_*	0	0	30	–	5.8 (32.0 *)	–

* The values in parentheses were obtained at the HF/6-31G(d) level [15]. ** The values in square parentheses are the *RE* corrected with zero-point energy at the MP2/6-31G(d) level.

**Table 2 molecules-28-02769-t002:** Five kinds of bond length (Å) of dodecahedrane analogues at the MP2/6-31G(d) level.

	C–C	Si–C	Si–Si	C–H	Si–H
** *a* **	1.58	1.89–1.92	2.34–2.35	1.10	1.50
** *b* **	1.57–1.58	1.89–1.93	2.34–2.35	1.10	1.49–1.50
** *c* **	1.58	1.88–1.92	2.34	1.10	1.50
** *d* **	1.58	1.90–1.92	2.35	1.10	1.50
** *e* **	1.56	1.90–1.92	2.36	1.10	1.50
** *f* **	1.58	1.96	2.30	1.10	1.49
** *g* **	1.55–1.57	1.94	2.32–2.37	1.10	1.49–1.50
C_20_H_20_	1.55, (1.55 *)	–	–	1.1 (1.1 *)	–
Cl^−^@Si_20_H_20_	–	–	2.36	–	1.51
Si_20_H_20_	–	–	2.36 (2.39 *)	–	1.50 (1.49 *)
C_20_H_20_ (expt.)	1.54	–	–	–	–
Cl^−^@Si_20_H_20_ (expt.)	–	–	2.35–2.36	–	1.40

* The values in parentheses were obtained at the HF/6-31G(d) level [15].

**Table 3 molecules-28-02769-t003:** Energy level of HOMO and LUMO and their energy gap of the Si/C equally mixed dodecahedrane isomers, dodecahedrane, and the silicon analogues at the MP2/6-31G(d) level. For the other molecules [31], see the text.

	HOMO/a.u.	LUMO/a.u.	HOMO–LUMO Gap /eV
* **a** *	−0.322	0.095	11.3
* **b** *	−0.325	0.094	11.4
* **c** *	−0.329	0.096	11.5
* **d** *	−0.332	0.093	11.6
* **e** *	−0.324	0.072	10.8
* **f** *	−0.264	0.084	9.5
* **g** *	−0.321	0.061	10.4
C_20_H_20_	−0.384	0.162	14.9
Si_20_H_20_	−0.334	0.034	10.0
C_30_H_40_ *	−0.364	0.165	14.4
Si_30_H_40_ *	−0.325	0.056	10.4
C_10_Si_20_H_40_ *	−0.345	0.076	11.5
C_20_Si_10_H_40_ *	−0.345	0.146	13.4

* The calculational level is MP2/cc-pVDZ [31].

## Data Availability

The data presented in this study are contained within the article and are also available in the Appendix A.

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
