# Peer review of "Theoretical Study of Si/C Equally Mixed Dodecahedrane Analogues"

_molecules, 2023, doi:10.3390/molecules28062769_

Round 1
Reviewer 1 Report
Manuscript id: molecules-2268312
Title: Theoretical study of the Si/C equally-mixed dodecahedrane analogues
In this work, the authors have examined various structural isomers of platonic organosilicon hydrocarbon, C10Si10H20. I have some reservations about publishing this work as such for the following reasons:
1. The authors mention that they have examined 1648 different isomers for C10Si10H20. I did have a look at the cross-reference [34; The Denumerants of Icosahedral Group for Symmetry Itemized Enumeration of Coisomeric Dodecahedrane Derivatives and Heteroanalogues] to get an idea about the total number of isomers and how that number was determined to be the total. The author of the cross-reference paper [34] talks about two different elemental compositions in his original article (in Table 2). One is C20H10X10. Another is C10H10X10. Let’s take X as element Si for this case. If so, then the elemental compositions will be C20H10Si10 in case one and C10H10Si10 in case two. Both are not equal to C10Si10H20 that the current authors have investigated. Having said that the authors need to give an explanation of how they got the total number of isomers for C10Si10H20 as equal to 1648?
2. The supplementary information submitted along with the original paper lacks a volume of information. No optimized Cartesian coordinates including electronic energies are given for any structure, which is a red flag in my opinion for a computational chemistry related paper. The authors must give this information for sure without any reservations if they want this paper to be published (whether in this journal or elsewhere). The authors did mention that they had carried out harmonic vibrational frequency calculations for all structures but I don’t see any information in the SI about the zero-point vibrational energies (ZPVE), ZPVE-corrected total energies, # of imaginary frequencies that they got for various structures, etc., Merely giving the jpeg image of the final structure or homo-lumo gaps and orbitals are not sufficient. It is also unclear how the authors have generated these many structures. Is it through manual permutations and combinations or through any search algorithm? Whatever be the case, the authors should give all the xyz coordinates as it will be a solid dataset that others could benefit in the future on these kinds of molecules.
3. Point 2 brings to relative energy that is mentioned in Table 1 of this paper. Are they (RE) ZPVE-corrected?
4. Isomers a, b, c, d, and e are within 1.1 kcal/mol at the MP2/6-31G(d) level. This makes me to question not only about the level that is applied here but also about the global minimum geometry. For these 5 isomers at least the authors must to some higher-level calculations like coupled-cluster to know the global minimum. If not then please do at least density functional theory calculations including empirical dispersion corrections for these five isomers.
I recommend a major revision for these reasons.
Minor corrections:
1. Line 81: wase determined?
Reviewer 2 Report
The effect of the number of Si-C bonds and strain energy on the relative energy of different configuration of the Si/C equally mixed dodecahedrane were investigated in this work. and the results were ell presented. However I wish to see more a bout the importance of this kind of materials in comparison to their similar counterpart. I see a quick note about the possibility of their application in the next generation of power semiconductor. I think it would be great if the authors could discuss how this materials could be utilized in the mentioned application in more details.
Round 2
Reviewer 1 Report
While the authors have addressed my questions and comments, the following points still need to be addressed so that this article can be accepted for publication.
1. Please provide xyz coordinates of the first 7 geometries in the Supplementary Material. This is still missing.
2. The format for Refs. 23, 24, and 37 do not seem to follow the format of others. For example, the year mentioned.
3. Refs. 28 and 29, some information is missing.
4. Please follow an uniform format for all references.
